# Molecular Microscope Diagnostic System in Patients after Kidney Transplantation—First Experience

**DOI:** 10.3390/biomedicines12030548

**Published:** 2024-02-29

**Authors:** Monika Beliančinová, Patrícia Kleinová, Tímea Blichová, Matej Vnučák, Karol Graňák, Katarína Kajová Macháleková, Ivana Dedinská

**Affiliations:** 1Transplant-Nephrology Department, University Hospital Martin, Kollárova 2, 036 59 Martin, Slovakia; pata.kleinova@gmail.com (P.K.); timea.blichova@gmail.com (T.B.); vnucak.matej@gmail.com (M.V.); granak.k@gmail.com (K.G.); ivana.dedinska@uniba.sk (I.D.); 2Department of Internal Medicine I, Jessenius Medical Faculty, Comenius University, 036 59 Martin, Slovakia; 3Department of Pathology, St. Elizabeth Cancer Institute Hospital, 811 08 Bratislava, Slovakia; katarina.machalekova@ousa.sk

**Keywords:** kidney transplantation, graft biopsy, rejection changes, molecular microscope, borderline

## Abstract

Background: The diagnosis of graft rejection relies on the identification of donor-specific antibodies along with histological findings. Borderline changes are particularly challenging, representing non-rejection findings in up to 70% of cases. The analysis aimed to compare the results of histopathological conclusions with the findings from examination using a molecular microscope, which assesses gene expression (whole-genome microarray chip technology). Methods: Molecular microscope examination (MMDx) was applied to twelve patients (six men and six women) who underwent either indication or protocol graft biopsy. Results: The average age of patients was 46.6 years ± 4.2 (average follow-up from kidney transplantation was 6.1 months ± 1.2). MMDx examination was performed during indication biopsy in 11 patients and protocol biopsy in 1 patient. A total of 33% of the findings matched and 50% did not. Finally, we present a case of a patient with acute cellular rejection findings without clinical and laboratory correlation, where the use of MMDx significantly altered the treatment strategy. Conclusions: MMDx examination is suitable for complementing patients with ambiguous histological findings and a clinical picture not corresponding to biopsy results. The limitations of MMDx include cost and its inability to evaluate the potential recurrence of the underlying kidney disease in the graft.

## 1. Introduction

The diagnosis of graft rejection is based on assessing the presence of de novo-generated donor-specific antibodies in combination with graft biopsy and evaluating the histological findings. The findings are set according to the criteria and terminology of the Banff classification, which are divided into six groups based on the presence/absence of rejection changes. Borderline findings form a distinct group, which are suspicious for acute rejection but can be non-rejection in up to 70% of cases. The findings include signs of tubulitis along with interstitial inflammation in up to 10% of the sample or the presence of tubulitis findings in a maximum of 4 cross-sections of tubules and simultaneously signs of interstitial inflammation in up to 25%, but no intimal arteritis is present. Non-rejection changes, described as borderline changes, may involve the findings of acute kidney injury (AKI) and inflammation. Tubulitis is a nonspecific sign that should not be evaluated if the tubules are atrophic, but inflammation is always present in atrophic tubules. Such evaluation requires experience in distinguishing whether the tubules are too atrophic for assessment. At present, scientists are trying to eliminate this category with the help of microarray studies. The molecular microscope provides the opportunity to objectively determine rejection and non-rejection changes in a histological sample labeled by a pathologist as borderline changes [1]. According to the Banff classification criteria, findings from groups 1–5 can coexist. The CERTPAP study dealt with a diverse description of the results of 55 biopsy samples of transplanted kidneys, which were sent to various centers with different pathologists, and it describes a significant interindividual variability in defining histopathological findings [2].

Molecular Microscopic Diagnostic System/Molecular Microscope (MMDx) operates by comparing the whole genome expression of biopsy samples using microchip technology in collaboration with artificial intelligence. mRNA is produced in transplanted kidneys when DNA gene expression is active. In injured or damaged organs, particular mRNA patterns are active. RNA is isolated from the biopsy sample and hybridized into a microarray. The microarray has a silicone chip with thousands of features printed with a specific oligonucleotide, which will hybridize with fragments in the biopsy mRNA. According to the sample database and machine learning, mRNA levels are analyzed and matched against the database, and the probability of rejection, acute damage, or fibrosis in the examined tissue is calculated. This process is a continual discovery and learning loop comparing every biopsy to the reference set. 

The molecular microscope is indicated in cases of inconclusive light microscopy results, such as borderline findings and transplant glomerulopathy, in correlation with the clinical picture and the pathologist’s findings or cases of insufficient or damaged material being used for examination in light microscopy. This approach helps overcome conventional biopsy diagnosis limitations based only on light microscopy and immunohistochemistry.

The analysis aimed to compare the results of histopathological conclusions of the examined samples using light microscopy with the findings of the examination using the molecular microscope and highlight the discordances between the two methods.

## 2. Materials and Methods

In a cohort of kidney transplant patients monitored at the Transplant-Nephrology Department of University Hospital Martin, who underwent either an indication or protocol graft biopsy, we identified those for whom examination by a molecular microscope was indicated.

We determined whether these cases involved an indication or protocol graft biopsy and compared the histopathological results with the results from MMDx.

At our institution, protocol biopsy is performed three months after kidney transplantation in all patients (except those under 18 years old). Graft biopsy is performed under local anesthesia, in aseptic conditions, in the procedure room for biopsies of transplanted kidneys. A 15G biopsy needle (VIGEO s.r.l., San Biagio di Bagnolo San Vito, Italy) is used to ensure an adequate sample thickness. Based on the macroscopic appearance of the sample, two to three samples are taken through one puncture. One to two pieces are preserved in the cold and separated for examination in light microscopy and immunohistochemical analysis. One sample, intended for MMDx examination, is held in an RNA stabilization solution. Before preservation in the stabilization solution, the sample must not be rinsed due to the risk of contamination and must be transferred to the solution using a sterile instrument. If the sample is not sent immediately, it is cooled to +4 degrees Celsius for sufficient fixative penetration into the tissue and then frozen at −20 °C 24 h after collection. Later, it is transported to the Molecular Medicine Laboratory at IKEM in Prague.

## 3. Results

A total of 12 patients were included in the study, for whom examination by a molecular microscope was indicated. MMDx examination was performed during an indication biopsy in 11 patients and a protocol biopsy in 1 patient. The average age of the patients was 46.4 years ± 4.2, and the average follow-up from kidney transplantation was 6.1 months ± 1.2. In four cases, the MMDx conclusion matched the pathologist’s conclusion of rejection but was discordant in terms of inflammation, atrophy, and fibrosis grade. Three samples were evaluated as rejection changes in light microscopy, but MMDx did not confirm the rejection. One sample was labeled as borderline changes, but MMDx excluded the presence of rejection. There were borderline changes in another case, but the MMDx examination discovered fully developed ABMR. The pathologist assessed one sample as non-representative for light microscopy, but examination using MMDx was possible, and no rejection was detected in MMDx. In another case, no rejection was detected with suspicion of the recurrence of FSGS, and MMDx discovered moderate to severe TCMR. In one case, there was insufficient material for a light microscope, so it was examined with MMDx only, with the discovery of mild molecular signs of TCMR. Table 1 provides a detailed specification of the findings and discordances in the results.

In the presented case study, we provide a perspective on a case of discrepancy, with the finding of acute cellular rejection IIA according to light microscopy showing focal positivity for C4d and a negative result for kidney transplant rejection according to MMDx.

## 4. Case Report

We describe the case of a 31-year-old patient with diabetic kidney disease due to type 1 diabetes mellitus who underwent primary kidney transplantation from a deceased donor at our facility in January 2022. The donor met standard criteria, and the patient was in a dialysis program for 522 days. The cold ischemia time was 18 h and 17 min. The patient was considered at low immunological risk based on immunological and non-immunological factors. Basiliximab (anti-CD25) + methylprednisolone 500 mg (D0 and D1) were chosen for induction. According to the protocol, maintenance immunosuppression was continued (tacrolimus, prednisone, mycophenolic acid). Post-transplantation, the graft showed primary function onset.

In the third month after kidney transplantation, the patient underwent a protocol biopsy of the graft, and a blood sample was taken to determine de novo donor-specific antibodies against the graft (dnDSA) using the Luminex method, which returned negative results. The histopathological analysis showed evidence of acute cellular rejection IIA according to the 2018 Banff classification, with focal positivity for C4d (Figure 1a–c). A specific immunohistochemistry method was used for C4d proof without pan-T antibody-antigen. Additionally, leukocyte infiltration was present, indicating a possible infection confirmed by urine culture, revealing significant bacteriuria (Klebsiella pneumoniae, 10^6^). This infection was treated explicitly with meropenem.

Considering the above, we initiated anti-rejection treatment in the patient. As the patient’s glycemic control was insufficient at that time, we decided on anti-rejection therapy using antithymocyte globulin (ATG). During the administration of the first dose of ATG, the patient developed an incipient cytokine release syndrome (CRS) with the onset of hypotension, fever, redness, and swelling in the facial area. Therefore, the ATG infusion was immediately stopped, and the patient responded well to the administered medication, including hydrocortisone, bisuleptin, paracetamol, and volume therapy.

Subsequently, the patient developed febrile episodes with a positive test for SARS-CoV-2, and we discontinued further ATG administration. Due to significant cytomegalovirus (CMV) replication (a total of 12 thousand copies/milliliter), we were even forced to reduce, and eventually wholly discontinue, mycophenolic acid therapy despite a biopsy-confirmed rejection that was left untreated.

Despite the histological findings, the patient maintained excellent graft function. After recovering from the COVID-19 infection and treating CMV viremia with valganciclovir, the patient had no clinical difficulties. Sonographic examination of the transplanted kidney also did not indicate severe rejection. Consequently, we decided to examine a biopsy sample using MMDx, revealing findings of interstitial fibrosis and tubular atrophy (IF/TA) grade 1 (Figure 2). Graft survival in similar cases reached 88% in the first year and 71% in the third year.

## 5. Discussion

In our analysis, 25% of the findings were consistent with light microscopy, and 50% differed. A study by authors from Dallas compared the results from endomyocardial biopsy light microscopy with MMDx, where the molecular microscope results were 94% concordant, with one case being different. The remaining divergent case involved cell-mediated graft rejection (TCMR) detection in the MMDx findings, while no signs of rejection were present in light microscopy [3]. The INTERCOMEX study, which compiled results from 519 biopsy samples from 10 European and American centers, determined a 77% agreement between MMDx and light microscopy in cases of TCMR, 77% in cases of ABMR, and a result without rejection findings in 76% of cases. The authors noted that the MMDx result correlated more with clinical judgment than with histological findings (87% vs. 80%; *p* = 0.0042) [4].

In our analysis, 17% of findings were described as rejection changes in light microscopy, but molecular microscopy excluded the presence of rejection. The discrepancy between MMDx and light microscopy, denying the light microscopy findings, is uncommon in our patients but not unprecedented. A case report by Lawrence et al. discusses a histopathological finding in a kidney transplant patient on maintenance therapy with tacrolimus, azathioprine, and prednisone. Six years post-transplant, nitrogenous waste retention rose (from 100 μmol/L to 153 μmol/L), while protein waste remained negative, and sonographic findings were normal. The histopathological finding corresponded to TCMR, Banff classification stage Ia, but MMDx rejected the presence of TCMR and ABMR, indicating the presence of IF/TA2 and AKI. Classical light microscopy findings were considered for graft function deterioration, and immunosuppressive therapy was changed. Due to the squamous cell carcinoma findings, antimetabolites (azathioprine, 6-mercaptopurine precursor) were not increased, and tacrolimus was not increased for arterial hyalinosis findings. Instead, the prednisone dose was raised to 30 mg daily and gradually tapered to a chronic dose of 5 mg daily after three months. Graft function did not improve, so a biopsy was indicated. This time, the light microscopy sample showed focal acute tubular injury, 10% IFTA without significant interstitial inflammation, and tubulitis with moderate arteriolar hyalinosis. The MMDx result was consistent with the previous finding [5]. A study by Swiss authors also compared the light microscopy results from graft biopsies with molecular microscopy. Overall, they reached 51 samples, with 18 showing discrepancies in the findings. Fifteen findings (83.3%) were described as suspicious for rejection according to light microscopy, but MMDx excluded the presence of rejection. In three (16.6%) findings, rejection signs were not described in light microscopy, but MMDx confirmed rejection signs [6]. Another two studies refer to discrepancies between light microscopy findings and molecular microscope examination at 37%. Reeve et al. collected data from 1208 indicated transplanted kidney biopsies from 13 centers with a goal of new molecular classification in kidney transplant biopsies. Authors reported 32% disagreement. One hundred and nine biopsies were suspected of borderline changes (TCMR lesions below the threshold required for canonical TCMR). Madil-Thomsen’s report showed disagreement in 37% out of the 1679 transplanted biopsy findings, with 20% showing no detected rejection with MMDx. Ambiguous histology findings, such as suspicious ABMR, transplant glomerulopathy, and borderline changes, showed high discrepancies with MMDx [7,8].

## 6. Conclusions

Despite the advanced technology that significantly aids in diagnosing and selecting the correct therapeutic approach for kidney transplant patients, it is crucial to consider that artificial intelligence should not serve as a replacement for the experience of histopathologists but as a sophisticated complement. It should be used with consideration of light microscopy, the immunohistochemical examination results, and an assessment of the patient’s clinical picture. Its utility is mainly found in cases of ambiguous light microscopy results, such as borderline and transplant glomerulopathy, in correlation with the patient’s clinical presence or cases where there is insufficient material for examination in light microscopy. The proper identification of patients for MMDx examination contributes to improving the diagnosis and treatment of findings that might otherwise go untreated under normal circumstances, potentially leading to a deterioration of transplanted kidney function.

## Figures and Tables

**Figure 1 biomedicines-12-00548-f001:**
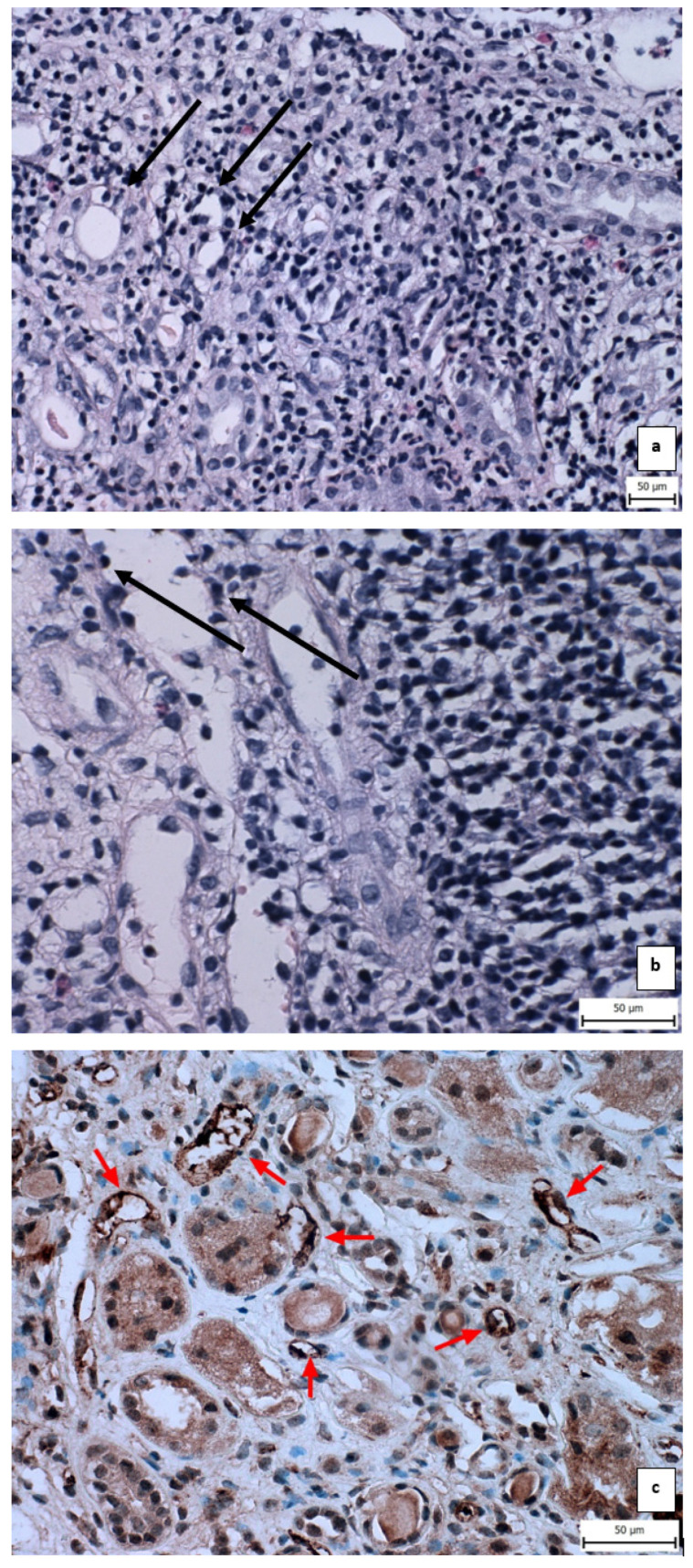
Histopathology findings from the biopsy sample of the presented case. (**a**): hematoxylin-eosin, ×200, detail with rejection infiltrate with leucocytes and tubulitis, black arrows point out leukocyte infiltration around the tubules; (**b**): hematoxylin-eosin, ×400, detail with venulitis, black arrows point out leukocyte infiltration around venules; (**c**): immunohistochemistry, ×400, detail with C4d positivity, red arrows point out C4d positivity around peritubular capillaries. Changed and retrieved from archive Kajová Macháleková, K.

**Figure 2 biomedicines-12-00548-f002:**
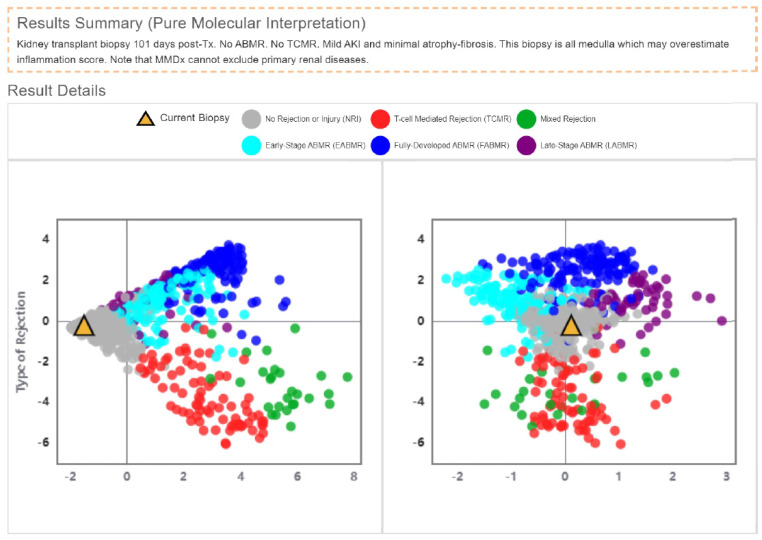
MMDx finding of the biopsy sample of the presented patient. No ABMR, no TCMR, mild AKI, and minimal atrophy−fibrosis. This biopsy is all medulla, which may overestimate the inflammation score. Changed and retrieved from archive Laboratory of Molecular Medicine, Laboratory of Medicine Genetics, Institute of Clinical and Experimental Medicine IKEM.

**Table 1 biomedicines-12-00548-t001:** Comparison of light microscopy and MMDx findings in biopsy of transplanted kidney.

Patient’s Credentials	Type of the Biopsy	Light Microscopy Findings	MMDX	Concordances/ Discordances
J.B.	I	Chronic ABMR, transplant glomerulopathy, FSGS	Severe, fully developed ABMR, moderate IFTA, and mild AKI	Concordant in rejection Fibrosis discordant
J.Š.	I	No rejection, discrete CNI toxicity, IFTA1	No ABMR/TCMR, IFTA1, minimal AKI and minimal inflammation	Concordant in rejection Inflammation discordant
T.D.	I	No rejection, transplantation glomerulopathy with FSGS, IFTA 2	No ABMR/no TCMR, AKI, IFTA 2	Concordant in rejection AKI discordance
D.O.	I	Acute ABMR	Mild, early-stage ABMR, no TCMR, extensive atrophy-fibrosis, moderate AKI, and inflammation	Concordant in rejection IFTA, AKI, and inflammation discordant
M.P.	P	Borderline changes	No ABMR/TCMR, minor molecular signs of ABMR, moderate inflammation, and IFTA1	Rejection discordance Inflammatory and IFTA discordance
S.H.	I	Borderline, transplantation glomerulopathy, ATN-like, IFTA 1	Fully developed ABMR, IFTA 3	Rejection discordance IFTA discordance
J.M.	I	TCMR IIa, C4d +	No ABMR/no TCMR, mild AKI, and minimal IFTA	Rejection discordance AKI and IFTA discordance
P.B.	I	Possible TCMR	No ABMR/TCMR, minimal AKI, minimal IFTA	Rejection, inflammation, and fibrosis discordant
M.B.	I	Suspect subclinical TCMR, possible infection injury	No ABMR/no TCMR	
M.D.	I	No rejection, IFTA2, possible recurrence of FSGF	Moderate to severe TCMR, no ABMR, Extensive atrophy and fibrosis, AKI gr.2	Rejection discordance
V.G	I	Not examined in light microscopy	No ABMR/no TCMR, mild molecular signs of TCMR, mild atrophy-fibrosis signs	Not applicable
Ľ.K.	I	Not representative, possible C4d focal positivity	No ABMR/no TCMR, mild AKI, minimal IFTA	Rejection discordance

ABMR—antibody-mediated rejection; AKI—acute kidney injury; CNI—calcineurin inhibitors; FSGS—focal segmental glomerulosclerosis; I—indication biopsy; IFTA—interstitial fibrosis and tubular atrophy; MMDx—molecular microscope; P—protocolar biopsy; TCMR—T-cell-mediated rejection.

## Data Availability

The original contributions presented in the study are included in the article, further inquiries can be directed to the corresponding author.

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
