# Peer review of "Molecular Microscope Diagnostic System in Patients after Kidney Transplantation—First Experience"

_biomedicines, 2024, doi:10.3390/biomedicines12030548_

Round 1
Reviewer 1 Report (Previous Reviewer 1)
Comments and Suggestions for Authors
Dear Authors
The resubmuttied manuscript entitled " USE OF THE MOLECULAR MICROSCOPE DIAGNOSTIC SYSTEM IN PATIENTS AFTER KIDNEY TRANSPLANTATION – FIRST EXPERIENCE" describes the use of the microscopic examination to assess patients after kidney transplantation.
The authors have succesfully addressed my initial comments, hence i do not feel that i have something to comment more. I think the manuscript has been significantly improved the performed revisions. Only minor comment i have
1) Please check the whole manuscript if exist any grammar or phrase errors.
2) I would suggest the change of the initial title of the manuscript.
Comments on the Quality of English LanguageThe english are fine, no issues detected
Author Response
|
Response to the first reviewer's Comments
|
||
|
1. Summary |
|
|
|
Thank you very much for taking the time to review this manuscript. Please find the detailed responses below and the corresponding revisions in the re-submitted files.
|
||
|
2. Questions for General Evaluation |
Reviewer’s Evaluation |
Response and Revisions |
|
Does the introduction provide sufficient background and include all relevant references? |
Yes/Can be improved/Must be improved/Not applicable |
|
|
Are all the cited references relevant to the research? |
Yes/Can be improved/Must be improved/Not applicable |
|
|
Is the research design appropriate? |
Yes/Can be improved/Must be improved/Not applicable |
|
|
Are the methods adequately described? |
Yes/Can be improved/Must be improved/Not applicable |
|
|
Are the results clearly presented? |
Yes/Can be improved/Must be improved/Not applicable |
|
|
Are the conclusions supported by the results? |
Yes/Can be improved/Must be improved/Not applicable
|
|
|
3. Point-by-point response to Comments and Suggestions for Authors |
||
|
Comments 1: Please check the whole manuscript if exist any grammar or phrase errors |
||
|
Response 1: Thank you for your comment. We checked the whole manuscript for any grammar or phrase errors.
|
||
|
Comments 2: I would suggest the change of the initial title of the manuscript. |
||
|
Response 2: Thank you for your comment. We agree, and therefore, we changed the title of the manuscript.
|
||
|
4. Response to Comments on the Quality of English Language |
||
|
Point 1: The english are fine, no issues detected |
||
|
Response 1: Thank you for your comment |
||
|
|
||
|
|
||
Reviewer 2 Report (Previous Reviewer 2)
Comments and Suggestions for Authors
In table I: patient V.G. histological data is not reported.
In Table I, insert a column in which the results of the concordance/discordance between the two techniques are clearly reported.
The quality of Figures 1 and 2 is low.
Figures 1 and 2, caption and labels. As to Figure 1, in all reference centers for the evaluation of transplant biopsies the evaluation of the lymphocytic and monocytic infiltrate is performed in immunohistochemistry using a pan-T antibody against the CD3 antigen and anti-macrophage against the CD68 antigen ; hematoxylin-eosin is not sufficient. Add a comment about it.
The term leukocytes is generic and non-specific; use the appropriate terms, such as lymphocytes, monocytes or other. The labels are incorrect and should be replaced by a), b) and c) and each image should be described with that reference in the caption. Figure No 2 does not show arteries but venous vessels; the correct term to describe the inflammation is endothelialitis and the picture illustrated is venulitis. The immunohistochemistry figure is of low technical quality; there is a strong background (false positive) at the level of the nuclei of the tubular epithelium, the cytoplasm of the tubular epithelium and the protein cylinders.
Insert a caption comment on the figure showing the data received from the MMDx center.
In any case, there are only two figures in the work, 1 (a/b/c) and 2 (MMDx); please, change according to the suggestion.
Author Response
|
Response to the second reviewer's Comments
|
||
|
1. Summary |
|
|
|
Thank you very much for taking the time to review this manuscript. Please find the detailed responses below and the corresponding revisions in the re-submitted files.
|
||
|
2. Questions for General Evaluation |
Reviewer’s Evaluation |
Response and Revisions |
|
Does the introduction provide sufficient background and include all relevant references? |
Yes/Can be improved/Must be improved/Not applicable |
|
|
Are all the cited references relevant to the research? |
Yes/Can be improved/Must be improved/Not applicable |
|
|
Is the research design appropriate? |
Yes/Can be improved/Must be improved/Not applicable |
|
|
Are the methods adequately described? |
Yes/Can be improved/Must be improved/Not applicable |
|
|
Are the results clearly presented? |
Yes/Can be improved/Must be improved/Not applicable |
|
|
Are the conclusions supported by the results? |
Yes/Can be improved/Must be improved/Not applicable
|
|
|
3. Point-by-point response to Comments and Suggestions for Authors |
||
|
Comments 1: In table I: patient V.G. histological data is not reported. |
||
|
Response 1: Thank you for your comment. Histological data of patient V.G. is not reported due to the patient's poor tolerance of the biopsy, and only tissue samples were taken for MMDx examination. We added information about light microscopy findings in Table No.1 as not examined and other patients examined at the time of the resubmission. |
||
|
Comments 2: In Table I, insert a column in which the results of the concordance/discordance between the two techniques are clearly reported. |
||
|
Response 2: Thank you for your comment. We added a third column with concordance/dis-concordance between the two techniques
Comments 3: The quality of Figures 1 and 2 is low. Response 3: Thank you for your comment. Unfortunately, the quality of figures a) and b) are of the best quality we can obtain from the pathologist. Original Figures can be seen in the uploaded pdf file.
Comments 4: Figures 1 and 2, caption and labels. As to Figure 1, in all reference centers for the evaluation of transplant biopsies the evaluation of the lymphocytic and monocytic infiltrate is performed in immunohistochemistry using a pan-T antibody against the CD3 antigen and anti-macrophage against the CD68 antigen ; hematoxylin-eosin is not sufficient. Add a comment about it Response 4: Thank you for your comment, The caption and labels have been corrected. In the Pathology Institute for evaluating transplanted kidney biopsies, pan-T antibody is not routinely used. You can see the added comment on the sixth page, line 144.
Comments 5: The term leukocytes is generic and non-specific; use the appropriate terms, such as lymphocytes, monocytes or other. The labels are incorrect and should be replaced by a), b) and c) and each image should be described with that reference in the caption. Figure No 2 does not show arteries but venous vessels; the correct term to describe the inflammation is endothelialitis and the picture illustrated is venulitis. The immunohistochemistry figure is of low technical quality; there is a strong background (false positive) at the level of the nuclei of the tubular epithelium, the cytoplasm of the tubular epithelium and the protein cylinders. Response 5: Thank you for the comment. The labels of figures were replaced, and captions were changed.
Comments 6: Insert a caption comment on the figure showing the data received from the MMDx center. Response 6: Thank you for the comment. The caption comment of the MMDx figure was changed
Comments 7: In any case, there are only two figures in the work, 1 (a/b/c) and 2 (MMDx); please, change according to the suggestion. Response 7: Thank you for the comment, we changed labels of the figures as you reccomended.
|
||
|
4. Response to Comments on the Quality of English Language |
||
|
N/A |
||
Reviewer 3 Report (Previous Reviewer 3)
Comments and Suggestions for Authors
The authors have addressed the concerns raised in the previous version. I have no further comments.
Author Response
|
Response to the third reviewer's Comments
|
||
|
1. Summary |
|
|
|
Thank you very much for taking the time to review this manuscript. Please find the detailed responses below and the corresponding revisions in the re-submitted files.
|
||
|
2. Questions for General Evaluation |
Reviewer’s Evaluation |
Response and Revisions |
|
Does the introduction provide sufficient background and include all relevant references? |
Yes/Can be improved/Must be improved/Not applicable |
|
|
Are all the cited references relevant to the research? |
Yes/Can be improved/Must be improved/Not applicable |
|
|
Is the research design appropriate? |
Yes/Can be improved/Must be improved/Not applicable |
|
|
Are the methods adequately described? |
Yes/Can be improved/Must be improved/Not applicable |
|
|
Are the results clearly presented? |
Yes/Can be improved/Must be improved/Not applicable |
|
|
Are the conclusions supported by the results? |
Yes/Can be improved/Must be improved/Not applicable
|
|
|
3. Point-by-point response to Comments and Suggestions for Authors |
||
|
Comments 1: The authors have addressed the concerns raised in the previous version. I have no further comments. Response 1: Thank you for the comment. |
||
|
4. Response to Comments on the Quality of English Language |
||
|
N/A |
||
Reviewer 4 Report (Previous Reviewer 4)
Comments and Suggestions for Authors
The paper titled "USE OF THE MOLECULAR MICROSCOPE DIAGNOSTIC SYSTEM IN PATIENTS AFTER KIDNEY TRANSPLANTATION – FIRST EXPERIENCE" presents interesting results regarding the use of Molecular microscope examination (MMDx) as a suitable complementing diagnostic tool for patients with ambiguous histological findings and a clinical picture not corresponding to biopsy results. The paper can be accepted in its present form.
Author Response
|
Response to the fourth reviewer's Comments
|
||
|
1. Summary |
|
|
|
Thank you very much for taking the time to review this manuscript. Please find the detailed responses below and the corresponding revisions in the re-submitted files.
|
||
|
2. Questions for General Evaluation |
Reviewer’s Evaluation |
Response and Revisions |
|
Does the introduction provide sufficient background and include all relevant references? |
Yes/Can be improved/Must be improved/Not applicable |
|
|
Are all the cited references relevant to the research? |
Yes/Can be improved/Must be improved/Not applicable |
|
|
Is the research design appropriate? |
Yes/Can be improved/Must be improved/Not applicable |
|
|
Are the methods adequately described? |
Yes/Can be improved/Must be improved/Not applicable |
|
|
Are the results clearly presented? |
Yes/Can be improved/Must be improved/Not applicable |
|
|
Are the conclusions supported by the results? |
Yes/Can be improved/Must be improved/Not applicable
|
|
|
3. Point-by-point response to Comments and Suggestions for Authors |
||
|
Comments 1: The paper titled "USE OF THE MOLECULAR MICROSCOPE DIAGNOSTIC SYSTEM IN PATIENTS AFTER KIDNEY TRANSPLANTATION – FIRST EXPERIENCE" presents interesting results regarding the use of Molecular microscope examination (MMDx) as a suitable complementing diagnostic tool for patients with ambiguous histological findings and a clinical picture not corresponding to biopsy results. The paper can be accepted in its present form. Response 1: Thank you for the comment. |
||
|
4. Response to Comments on the Quality of English Language |
||
|
N/A |
||
Round 2
Reviewer 2 Report (Previous Reviewer 2)
Comments and Suggestions for Authors
Please check again Figure 1 legend; it should be like this:
Figure 1. Histopathology findings from the biopsy sample of the presented case. a): Hematoxylin-eosin, x200, detail with rejection infiltrate with leucocytes and tubulitis; b): Hematoxylin-eosin, x400, detail with venulitis; c): Immunohistochemistry, x400, detail with C4d positivity.
Consequently, the next image should be marked as Figure 2 and not 3.
Lines 145-146. Please take into consideration the new numering of the images; The histopathological analysis showed evidence of acute cellular rejection IIA according to the 2018 Banff classification, with focal positivity for C4d (Figures 1a, b, c).
Author Response
Please see the attachment.

This manuscript is a resubmission of an earlier submission. The following is a list of the peer review reports and author responses from that submission.
Round 1
Reviewer 1 Report
Comments and Suggestions for Authors
Dear Authors,
The manuscript entitled " USE OF THE MOLECULAR MICROSCOPE DIAGNOSTIC SYSTEM IN PATIENTS AFTER KIDNEY TRANSPLANTATION – FIRST EXPERIENCE" describes the use of the microscopic examination to assess patients after kidney transplantation. I would suggest major comments for the submitted manuscript.
1) Have the authors confirmed their results with other approaches such as DSA.
2) Have the authors considered to use their approach in combination, with cf-DNA determination,
3) Original magnification and scale bars are required in figure 1. Also, the authors are mentioning that using Hematoxylin and Eosin stain. However, the stain looks different from the regular images of H&E stain. Also, background correction is required. The authors can also use arrows in the figure 1 in order to show better the infiltration of leukocytes.
4) Besides the H&E stain, have the authors considered to use immunohistochemistry approaches in the same tissue biopsis, for determining more precisely the CD4+, CD19+, CD11b+ or other populations.
Author Response
|
Response for the first reviewer
|
||
|
1. Summary |
|
|
|
Thank you very much for taking the time to review this manuscript. Please find the detailed responses below and the corresponding revisions in the re-submitted files.
|
||
|
2. Questions for General Evaluation |
Reviewer’s Evaluation |
Response and Revisions |
|
Does the introduction provide sufficient background and include all relevant references? |
Yes/Can be improved/Must be improved/Not applicable |
|
|
Are all the cited references relevant to the research? |
Yes/Can be improved/Must be improved/Not applicable |
|
|
Is the research design appropriate? |
Yes/Can be improved/Must be improved/Not applicable |
|
|
Are the methods adequately described? |
Yes/Can be improved/Must be improved/Not applicable |
|
|
Are the results clearly presented? |
Yes/Can be improved/Must be improved/Not applicable |
|
|
Are the conclusions supported by the results? |
Yes/Can be improved/Must be improved/Not applicable
|
|
|
3. Point-by-point response to Comments and Suggestions for Authors
|
||
|
Comment 1: Have the authors confirmed their results with other approaches such as DSA. |
||
|
Response 1: Thank you for your question. We have confirmed our results with no circulating DSA detected in the screening as mentioned in the manuscript (page 4, line 125-127) “..blood sample was taken to determine de novo donor-specific antibodies against the graft (dnDSA) using the Luminex method, which returned negative results.”. For all subjects, we confirmed DSA present in ABMR cases, and in other cases, we confirmed there are no circulating DSA (TCMR or no rejection cases).
|
||
|
Comment 2: Have the authors considered to use their approach in combination, with cf-DNA determination |
||
|
Response 2: Thank you for the comment. No, we haven´t; nowadays, cf-DNA is only sometimes used for examination and clarification of the diagnosis and is not routinely used in every transplant center. Even the largest transplant center in Europe (IKEM, Prague, Czech republic) doesn´t perform cfDNA determination.
Comment 3: Original magnification and scale bars are required in Figure 1. Also, the authors are mentioning that using Hematoxylin and Eosin stain. However, the color looks different from the regular images of H&E stain. Also, background correction is required. The authors can also use arrows in figure 1 to show better the infiltration of leukocytes. Response 3: Thank you for your comment. As for Figure 1, we have added information about the original magnification and scale bars in the revised manuscript. We are aware that it doesn´t look like Hematoxylin and Eosin stain. However, this is due to the concentration of the color stain.
Comment 4: Besides the H&E stain, have the authors considered to use immunohistochemistry approaches in the same tissue biopsis, for determining more precisely the CD4+, CD19+, CD11b+ or other populations Response 4: Thank you for the comment. We have also done other approaches in the mentioned case report, but not immunohistochemistry for specific lymphocyte specification as it is not routinely used. For clarification of the diagnosis, the specific immunohistochemistry method was performed (with C4d positivity), which can be seen in the added image (page 3).
|
||
|
4. Response to Comments on the Quality of English Language |
||
|
Point 1: English language fine. No issues detected |
||
|
Response 1: Thank you for the comment
|
||
|
5. Additional clarifications |
||
|
none |
||
Reviewer 2 Report
Comments and Suggestions for Authors
Diagnoses should be reviewed by sending cases to a high-output center for a second opinion, as such low agreement (25%) is embarrassing.
The case presented as emblematic of this discrepancy is of dubious evaluation; the histological images are of low quality, the percentage of T lymphocyte infiltration should be highlighted using a specific immunohistochemical method (anti-CD3) and there is no evidence of the anti CD4d reported in the text and not shown in figures 1-2.
Finally, the case is complicated by multiple bacterial and viral infections (Covid-19 and CMV) which may have caused an incorrect interpretation of the case at a histological level.
Author Response
|
Response for the second reviewer
|
||
|
1. Summary |
|
|
|
Thank you very much for taking the time to review this manuscript. Please find the detailed responses below and the corresponding revisions in the re-submitted files.
|
||
|
2. Questions for General Evaluation |
Reviewer’s Evaluation |
Response and Revisions |
|
Does the introduction provide sufficient background and include all relevant references? |
Yes/Can be improved/Must be improved/Not applicable |
|
|
Are all the cited references relevant to the research? |
Yes/Can be improved/Must be improved/Not applicable |
|
|
Is the research design appropriate? |
Yes/Can be improved/Must be improved/Not applicable |
|
|
Are the methods adequately described? |
Yes/Can be improved/Must be improved/Not applicable |
|
|
Are the results clearly presented? |
Yes/Can be improved/Must be improved/Not applicable |
|
|
Are the conclusions supported by the results? |
Yes/Can be improved/Must be improved/Not applicable
|
|
|
3. Point-by-point response to Comments and Suggestions for Authors |
||
|
Comment 1: Diagnoses should be reviewed by sending cases to a high-output center for a second opinion, as such low agreement (25%) is embarrassing. |
||
|
Response 1: Thank you for your comment. The aim of this study is to present only discrepant findings in light microscopy and molecular microscope. Only unclear and ambiguous histological findings are sent for MMDx examination; most cases have corresponding histological, laboratory, and clinical findings, which are not included in the study.
|
||
|
Comment 2: The case presented as emblematic of this discrepancy is of dubious evaluation; the histological images are of low quality, the percentage of T lymphocyte infiltration should be highlighted using a specific immunohistochemical method (anti-CD3) and there is no evidence of the anti CD4d reported in the text and not shown in figures 1-2. |
||
|
Response 2: Thank you for the comment. The MMDx examination of the samples is used only in cases with discrepant clinical, laboratory, and histological findings; MMDx is not used routinely in every kidney biopsy. The histological images were updated to the highest quality possible. Specific immunohistochemical methods are not routinely used for lymphocyte specification. There was added figure No.3 with immunohistochemistry C4d positivity.
Comment 3: Finally, the case is complicated by multiple bacterial and viral infections (Covid-19 and CMV) which may have caused an incorrect interpretation of the case at a histological level. Response 3: Thank you for your comment. You are right, the case is complicated by multiple bacterial and viral infections, which may have caused incorrect interpretation, but even the large meta-analysis which was aimed to find solid organ rejection after SARS-CoV-2 infection or vaccination mentions only 36 cases of transplanted kidney rejection out from 136 cases of solid organ rejections after SARS-CoV-2 infection. There were 56 cases of organ rejection after vaccination, with a median time of rejection total of 13,5-14 hours after vaccination. As you can see, we added Figure No.3, where C4d positivity can be clearly seen, so we didn´t think about SARS-Cov-2 infection related rejection as major cause of the rejection in presented case. Because of the possibility of incorrect interpretation, we examined the sample with MMDx.
|
||
|
4. Response to Comments on the Quality of English Language |
||
|
Point 1: I am not qualified to assess the quality of English in this paper |
||
|
Response 1: Thank you for the comment. |
||
|
5. Additional clarifications |
||
|
none |
||
Reviewer 3 Report
Comments and Suggestions for Authors
Beliančinová et al. present a data set from 12 patients after kidney transplant and one additional patient after acute cellular rejection without clinical or laboratory correlation. The authors compared molecular microscope examination (MMDx) which assesses gene expression (whole-genome microarray chip technology) with standard measures of graft rejection.
The findings between the methods was poor, with 25% matching and 50% not. One of the final conclusions was that MMDx could assist with the assessment of patients with ambiguous findings.
Aside from not really convincing this reviewer that MMDx is particularly helpful with this small cohort of patients, there are ethical problems.
The participants gave their consent. They did not give their written consent, and this prospective investigation did not have Institutional Review Board approval and oversight. This is not a case report, but is a work involving 13 people having their medical information presented publicly without human protections.
I have no further comments.
Author Response
|
Response for the third reviewer
|
||
|
1. Summary |
|
|
|
Thank you very much for taking the time to review this manuscript. Please find the detailed responses below and the corresponding corrections in the re-submitted files.
|
||
|
2. Questions for General Evaluation |
Reviewer’s Evaluation |
Response and Revisions |
|
Does the introduction provide sufficient background and include all relevant references? |
Yes/Can be improved/Must be improved/Not applicable |
|
|
Are all the cited references relevant to the research? |
Yes/Can be improved/Must be improved/Not applicable |
|
|
Is the research design appropriate? |
Yes/Can be improved/Must be improved/Not applicable |
|
|
Are the methods adequately described? |
Yes/Can be improved/Must be improved/Not applicable |
|
|
Are the results clearly presented? |
Yes/Can be improved/Must be improved/Not applicable |
|
|
Are the conclusions supported by the results? |
Yes/Can be improved/Must be improved/Not applicable
|
|
|
3. Point-by-point response to Comments and Suggestions for Authors |
||
|
Comment 1: The findings between the methods was poor, with 25% matching and 50% not. One of the final conclusions was that MMDx could assist with the assessment of patients with ambiguous findings. |
||
|
Response 1: Thank you for your comment. Only ambiguous histological findings are sent for MMDx examination; most cases have corresponding histological, laboratory, and clinical findings and these are not included in the study.
|
||
|
Comment 2: Aside from not really convincing this reviewer that MMDx is particularly helpful with this small cohort of patients, there are ethical problems. |
||
|
Response 2: The MMDx examination of the samples is used only in a small number of cases with discrepant clinical, laboratory, and histological findings; MMDx is not used routinely in every kidney biopsy. Every patient signed a written consent with participation in this study and authors followed rules by the Declaration of Helsinki of 1975 with revision in 2013.
Comment 3: The participants gave their consent. They did not give their written consent, and this prospective investigation did not have Institutional Review Board approval and oversight. This is not a case report, but is a work involving 13 people having their medical information presented publicly without human protections. Response 3: Thank you for your comment. We are sorry for the incomplete information; all of the patients gave their written consent with a kidney biopsy, examination of the biopsy sample, and presentation of their case. Also, the presentation of cases was approved by our local Ethical Board Committee with rules by the Declaration of Helsinki of 1975 with revision in 2013.
|
||
|
4. Response to Comments on the Quality of English Language |
||
|
Point 1: I am not qualified to assess the quality of English in this paper |
||
|
Response 1: Thank you for the comment |
||
|
5. Additional clarifications |
||
|
none |
||
Reviewer 4 Report
Comments and Suggestions for Authors
The paper titled "USE OF THE MOLECULAR MICROSCOPE DIAGNOSTIC SYSTEM IN PATIENTS AFTER KIDNEY TRANSPLANTATION – FIRST EXPERIENCE" presents interesting results regarding the use of Molecular microscope examination (MMDx) as a suitable complementing diagnostic tool for patients with ambiguous histological findings and a clinical picture not corresponding to biopsy results. The paper can be published after minor English revision.
Comments on the Quality of English LanguageEnglish proofreading is highly recommended
Author Response
|
Response for the fourth reviewer
|
||
|
1. Summary |
|
|
|
Thank you very much for taking the time to review this manuscript. Please find the detailed responses below and the corresponding corrections in the re-submitted files.
|
||
|
2. Questions for General Evaluation |
Reviewer’s Evaluation |
Response and Revisions |
|
Does the introduction provide sufficient background and include all relevant references? |
Yes/Can be improved/Must be improved/Not applicable |
|
|
Are all the cited references relevant to the research? |
Yes/Can be improved/Must be improved/Not applicable |
|
|
Is the research design appropriate? |
Yes/Can be improved/Must be improved/Not applicable |
|
|
Are the methods adequately described? |
Yes/Can be improved/Must be improved/Not applicable |
|
|
Are the results clearly presented? |
Yes/Can be improved/Must be improved/Not applicable |
|
|
Are the conclusions supported by the results? |
Yes/Can be improved/Must be improved/Not applicable
|
|
|
3. Point-by-point response to Comments and Suggestions for Authors |
||
|
The paper titled "USE OF THE MOLECULAR MICROSCOPE DIAGNOSTIC SYSTEM IN PATIENTS AFTER KIDNEY TRANSPLANTATION – FIRST EXPERIENCE" presents interesting results regarding the use of Molecular microscope examination (MMDx) as a suitable complementing diagnostic tool for patients with ambiguous histological findings and a clinical picture not corresponding to biopsy results. The paper can be published after minor English revision. |
||
|
|
||
|
|
||
|
|
||
|
4. Response to Comments on the Quality of English Language |
||
|
Point 1: English proofreading is highly recommended |
||
|
Response 1: Thank you for the comment. The manuscript was proofread, minor revisions were done.
|
||
|
5. Additional clarifications |
||
|
none |
||